# Enhanced Electrophoretic Depletion of Sodium Dodecyl Sulfate with Methanol for Membrane Proteome Analysis by Mass Spectrometry

**DOI:** 10.3390/proteomes12010005

**Published:** 2024-02-02

**Authors:** Hammam H. Said, Alan A. Doucette

**Affiliations:** Department of Chemistry, Dalhousie University, 6274 Coburg Road, Halifax, NS B3H 4R2, Canada; hm679392@dal.ca

**Keywords:** membrane proteins, sodium dodecyl sulfate, protein purification, automated sample preparation, electrophoresis, mass spectrometry, methanol

## Abstract

Membrane proteins are underrepresented during proteome characterizations, primarily owing to their lower solubility. Sodium dodecyl sulfate (SDS) is favored to enhance protein solubility but interferes with downstream analysis by mass spectrometry. Here, we present an improved workflow for SDS depletion using transmembrane electrophoresis (TME) while retaining a higher recovery of membrane proteins. Though higher levels of organic solvent lower proteome solubility, we found that the inclusion of 40% methanol provided optimal solubility of membrane proteins, with 86% recovery relative to extraction with SDS. Incorporating 40% methanol during the electrophoretic depletion of SDS by TME also maximized membrane protein recovery. We further report that methanol accelerates the rate of detergent removal, allowing TME to deplete SDS below 100 ppm in under 3 min. This is attributed to a three-fold elevation in the critical micelle concentration (CMC) of SDS in the presence of methanol, combined with a reduction in the SDS to protein binding ratio in methanol (0.3 g SDS/g protein). MS analysis of membrane proteins isolated from the methanol-assisted workflow revealed enhanced proteome detection, particularly for proteins whose pI contributed a minimal net charge and therefore possessed reduced solubility in a purely aqueous solvent. This protocol presents a robust approach for the preparation of membrane proteins by maximizing their solubility in MS-compatible solvents, offering a tool to advance membrane proteome characterization.

## 1. Introduction

Mass spectrometry (MS) remains indispensable for in-depth proteome characterization studies. The analysis of membrane proteins holds particular significance, as their functional roles in cellular transport and recognition translate into potential targets for novel molecular therapeutics [1,2]. Despite their importance, membrane proteins are generally underrepresented in proteomics data due primarily to their lower solubility, which risks sample loss during front-end sample preparation [3,4]. Several detergent-based strategies have been explored to enhance the recovery of membrane proteins, including the use of ionic surfactants such as sodium dodecyl sulfate (SDS), non-ionic surfactants, zwitterionics, and acid-cleavable detergents [5,6,7,8,9]. These latter classes of detergents are considered MS-compatible, whereas ionic surfactants are not. Even traces of SDS (0.01%) can adversely impact enzyme activity, ref. [10] complicate chromatographic resolution, and suppress MS signals by interfering with electrospray ionization (ESI) [11]. Nonetheless, SDS is recognized as the favored detergent to maximize cell lysis efficiency [12], proteome extraction [13], and protein solubilization [14].

As an alternative to surfactants, organic solvents have not only been shown to improve membrane protein solubility [15,16] but have also been suggested to enhance trypsin digestion activity [17]. For example, Blonder et al. reported that the addition of 60% methanol provided optimal protein solubility [15]. Moreover, Park and Min showed that trypsin activity is enhanced with the addition of 30% methanol [17]. By contrast, previous work from our group and that of others has shown that higher concentrations of organic solvent will hinder enzyme activity [18,19]. Furthermore, elevated levels of organic solvent will induce protein precipitation [20,21,22]. As little as 50% ethanol or 50% methanol is routinely employed to initiate precipitation and removal of serum proteins [22], which is useful to eliminate protein interferences in large-scale metabolome workflows [23]. Consequently, the optimal solvent concentration to enhance membrane protein solubility remains to be determined.

While organic solvents offer a potential ‘MS-compatible’ solution for membrane proteome characterization, ionic surfactants are still employed as the gold standard approach to membrane proteome solubilization [7]. Multiple sample preparation workflows have described the removal of SDS ahead of bottom-up MS analysis [24]. These include semi-automated approaches to manipulating samples using cartridges or on beads such as Filter-Aided Sample Preparation (FASP), Suspension Trapping (S-Trap), Single Pot Solid Phase Sample Preparation (SP3), or the ProTrap XG, a two-stage filtration and extraction cartridge [25,26,27,28]. The recent study by Varnivades et al. [24] highlights a direct comparison of 16 sample preparation strategies for bottom-up proteomics, including multiple detergent-based methods. Strong agreement in the methods was demonstrated, wherein over 61% of the nearly 3000 identified proteins were common across all 16 preparations. However, the study compared sample preparation time (excluding digestion), noting that FASP required the longest preparation of all methods employed (4 h). The fastest approach they examined, aptly named SPEED (Sample Preparation by Easy Extraction and Digestion) [29] requires 1 h of preparation.

Achieving high throughput for an analytical workflow is a favorable objective in proteomics analysis. However, all optimized sample preparation approaches must also achieve the desired protein purity while maintaining high analyte recovery. FASP, for example, has proven to be highly effective in depleting SDS but suffers highly variable proteome recovery, with sample losses of 50% or more [30]. As an alternative, we introduced a fully automated approach to SDS depletion known as transmembrane electrophoresis (TME) [31,32,33]. TME employs an electric field applied perpendicular to a molecular weight cut-off (MWCO) membrane, which retains the protein while allowing dodecyl sulfate anions to migrate through the membrane and towards the anode (sodium in turn migrates towards the cathode, though this is less of a concern as a protein interference) [32]. The TME approach enables rapid and reproducible proteome purification, whereby SDS is exponentially depleted at a rate proportional to the magnitude of the electric field. Nonetheless, Joule heating presents a significant risk to protein loss, particularly following SDS depletion, where elevated temperatures can induce protein aggregation [31,32]. Joule heating also challenges the process of accelerated SDS removal; increasing the magnitude of the applied voltage would theoretically allow faster SDS depletion, though the resulting temperature increase would compromise protein recovery and eventually lead to the boiling of the sample solution [33]. The fastest TME depletion experiments reported to date employ an active cooling system to better manage Joule heating and maintain high protein recovery, but still require 5 min for SDS removal [33].

As a general strategy, we herein propose the incorporation of methanol into a TME depletion experiment, which serves to retain membrane protein solubility throughout the SDS depletion process. This organic solvent is compatible with electrophoresis, as well as the subsequent enzyme digestion step and ESI MS. Prior studies have speculated that the inclusion of methanol (20%) in the transfer buffer promotes the dissociation of SDS from proteins during Western blots [34,35]. As our findings show here, the addition of organic solvent to TME not only enhances the solubility of proteins from a membrane-enriched proteome fraction but also accelerates the rate of SDS depletion by TME. This optimized workflow significantly advances the potential to analyze membrane proteins by bottom-up mass spectrometry.

## 2. Materials and Methods

Bovine serum albumin (BSA) was purchased from Millipore Sigma (Oakville, ON, Canada). The Pierce BCA assay kit and HPLC-grade solvents were sourced from Thermo Fisher Scientific (Ottawa, ON, Canada). Milli-Q-grade water was purified to 18.2 MΩ cm. The yeast, *Saccharomyces cerevisiae*, was obtained as a dry pellet from the grocery store.

### 2.1. Membrane Protein Isolation

The yeast was cultured overnight (30 °C) in YPD broth to an OD_600_ of 1, according to standard procedures [36]. To isolate the membrane-enriched proteome fraction, freshly grown yeast cells were lysed via French press, then combined in a 1:1 ratio with a sodium carbonate buffer (pH 11) and incubated on ice for 1 h. This was followed by two rounds of ultracentrifugation at 115,000 g, as previously described [37]. The resulting membrane-enriched pellet was resuspended in Tris buffer (pH 8.1, 100 mM) containing 5% SDS to a final protein concentration of 2 g/L as measured by the BCA assay. Prior to TME loading for SDS depletion, the 2 g/L sample was diluted 10-fold with either water or a water-methanol mixture, resulting in a final working protein concentration of 0.2 g/L with 0.5% SDS.

### 2.2. Membrane Protein Resolubilization with Methanol

A visual overview of the membrane proteome resolubilization experiment is provided in Appendix A. The membrane-enriched pellet was resuspended in water by vigorously dispersing the solid pellet using a syringe. Equal volumes of the homogenized pellet were aliquoted to multiple vials, then centrifuged prior to removing and discarding the aqueous supernatant. This revealed multiple vials with equal quantities of the unsolubilized membrane proteome pellet. To assess the capacity of various methanol/water solvents (ranging from 0 to 60% methanol) for the resolubilization of proteins, 50 µL of each solvent system was added to triplicate vials. The control solvent consisted of 0.5% SDS in water. The sample was then vortexed for 1 min, followed by sonication for 15 min, and then 30 min incubation at room temperature. The vials were then centrifuged, and the resulting supernatant was retained for subsequent analysis by BCA assay to determine the relative protein recovery as well as for SDS depletion by transmembrane electrophoresis.

### 2.3. Transmembrane Electrophoresis

The custom-built TME device, shown in Appendix A, was assembled as described previously [33]. The device uses a 3.5 kDa regenerated cellulose dialysis membrane (Thermo Fisher Scientific) and is operated with Tris/Tricine buffer (63 mM Tris, 100 mM Tricine). Both the cathode and anode chambers had a volume of 48 mL. The device supplies 5 sample cells, each 3 mm in thickness and 1 cm in diameter, and can accommodate up to 250 µL of sample per cell. Unless otherwise stated, TME was operated in constant current mode (250 mA) with active cooling (AC) achieved by circulating cold water through glass channels that cool the TME buffer within the cathode and anode chambers [33]. The membrane-enriched proteome extracts, initially containing 0.5% SDS prepared in water, were loaded into each of the 5 sample cells of the TME device and subjected to SDS depletion at 250 mA for 9 min, at which point the still-soluble proteins were recovered from the device by pipetting the solution from the sample well. In a separate TME run, the membrane-enriched fraction, prepared in 0.5% SDS with 40% methanol, was subjected to 3 min SDS depletion at 250 mA, after which the soluble extract was recovered from the TME wells. Samples were then subjected to protein recovery (BCA), residual SDS (MBAS), and bottom-up proteomics processing (protein digestion and LC-MS/MS).

### 2.4. SDS Colorimetric Assay by MBAS

The methylene blue active substances (MBAS) assay was used to determine residual SDS following TME depletion [38]. For this, 100 µL of the sample was mixed with 100 µL of methylene blue reagent (250 µg methylene blue, 50 g sodium sulfate, and 10 mL concentrated sulfuric acid per liter of aqueous reagent). Next, 400 µL of chloroform was added, and the sample was briefly vortexed. The methylene blue-SDS complex, which extracts into the chloroform layer, was quantified by absorbance (652 nm) against a calibration curve consisting of SDS standards ranging from 5 to 25 ppm. To account for interference caused by methanol in the MBAS assay, all TME samples were first fully dried in a Speedvac, and subsequently redissolved in an equal volume of water. Samples were also diluted appropriately prior to the addition of the methylene blue reagent to ensure that the SDS concentration fell within the range of the calibration curve.

### 2.5. Protein Digestion

All yeast membrane-enriched protein samples underwent digestion using standard procedures. The digestion was preceded by a reduction in 5 mM (final) dithiothreitol and alkylation using 11 mM iodoacetamide, respectively. Subsequently, protein samples underwent overnight digestion at 37 °C using 2% (*w*/*w*) trypsin, following a previously established procedure [26]. Samples were then desalted on a self-packed Poros 20 R2 column with LC-UV to quantify the digested peptides, as described previously [39].

### 2.6. CMC Determination

Solutions of 2% SDS were prepared in water with variable quantities of methanol ranging from 0 to 40% by volume. These solutions were then dispensed from a burette in ~1 mL increments to a beaker containing 100 mL of the corresponding water/methanol solution without SDS. An Extech EC150 conductivity meter (Nashua, NH, USA) was placed in the 100 mL beaker and used to record the changing conductivity throughout the addition of the SDS-containing solution. The conductivity readings were plotted relative to the diluted SDS concentration to reveal a distinct break in the slope, corresponding to the critical micelle concentration [40]. All measurements were recorded at room temperature (21 ± 1 °C) and performed in triplicate. The methanol-containing samples were benchmarked relative to the CMC of SDS in water [41].

### 2.7. SDS—Protein Binding Ratio in Methanol

The SDS-to-protein binding ratio was determined by dialysis [41]. Solutions were prepared containing 2 g/L BSA, 20 g/L SDS, and 0.1 mol/L NaCl in water, or water with 40% methanol by volume. After 1 h preincubation, 5 mL of the sample was loaded into a dialysis tube with a molecular weight cutoff of 6000 Da. The sealed tube was immersed in 250 mL of solvent matching the sample, excluding SDS and BSA. The solvent was changed twice, at 24 h intervals. Finally, BCA and MBAS assays were conducted as described above to determine the final protein and SDS concentrations, respectively.

### 2.8. LC-MS/MS

Beginning with a single preparation of the enriched membrane proteome fraction of yeast, two technical replicates were prepared for SDS depletion by TME with 40% methanol, and two more technical replicates were employed for SDS depletion by conventional TME in water. Following SDS depletion and trypsin digestion, 1 µL from each of the 4 samples, representing 1/10th of the processed membrane-enriched proteome fraction, was introduced into a Dionex Ultimate 3000 LC nanosystem (Bannockburn, IL, USA) system connected to a LTQ Orbitrap Velos Pro mass spectrometer. The LC employed a self-packed C18 column coupled to a 10 μm New Objective PicoTip noncoated Emitter Tip (Woburn, MA, USA). The 2 h linear gradient started with 0.1% formic acid in water, progressing to 35% acetonitrile to facilitate peptide separation. The MS instrument operated in data-dependent mode, with MS1 at a resolution of 30,000 FWHM, while MS2 rapidly scanned at 66,666 Da s^–1^, with a resolution of less than 0.6 Da FWHM. Proteins were identified and profiled from duplicate injections.

### 2.9. Data Analysis

MS data were searched with Proteome Discoverer software, version 1.4 (Thermo Fisher Scientific), with parent ion tolerance of 10 ppm and fragment tolerance of 0.5 Da, against the UniProt *S. cerevisiae* filtered database (6735 entries, accessed on 15 February 2023), with a false discovery rate of 5% and 1 unique peptide per protein. The grand average of hydropathicity (GRAVY) scores were calculated from an online tool available at http://www.gravy-calculator.de/ (accessed on 15 February 2023). Protein charge calculations were performed based on the amino acid sequence and associated pKa of the side chain residues. Cellular components were determined using Gene Ontology and functional annotation provided by DAVID 2021 software (Database for Annotation, Visualization, and Integrated Discover), available at https://david.ncifcrf.gov/tools.jsp (accessed on 15 February 2023) [42,43]. The intensity ratio of peptides identified from the respective TME solvent systems was recorded as a measure of the relative abundance of the identified protein retained in each solvent system following SDS depletion.

## 3. Results

### 3.1. Enhanced Membrane Protein Recovery with 40% Methanol

While previous work employed 60% (*v*/*v*) methanol to assist in the solubilization of membrane proteins [15,16], our prior experience suggests that such a high level of organic solvent may lead to protein precipitation [20]. We therefore assessed various concentrations of methanol in water to maximize the extraction of proteins from a membrane-enriched proteome fraction. As seen in Figure 1A,B, the addition of 30 or 40% methanol provided the highest total protein recovery, with a relative yield of 84 to 86% compared to samples extracted with 0.5% SDS in water. In 50% methanol, recovery dropped significantly to 32.1 ± 1%. Increasing methanol to 60% further lowered the recovery of proteins. SDS-PAGE of the extracted proteins confirmed these results (Figure 1B).

We next tested the impact of adding methanol to an already-solubilized membrane-enriched proteome fraction, extracted with 0.5% SDS. From Figure 1C, diluting the sample to a final concentration of 50% methanol severely compromised protein solubility. We observed the formation of a visible protein pellet, confirming that the addition of 50% methanol had induced protein precipitation. By contrast, adding methanol to 40% preserved the solubility of all proteins in the SDS-containing extract.

While 40% methanol enhances the extraction of a membrane-enriched proteome fraction relative to water alone, our results also indicate that 0.5% SDS consistently recovers a greater concentration of protein. This justifies the use of a detergent-based workflow for maximum protein recovery. We next set out to deplete the SDS from the membrane proteome extract using TME. The critical value permitting subsequent MS analysis is 100 ppm. We therefore operated the TME system until SDS dropped below 100 ppm. A summary of results obtained from various TME experiments is provided in Table 1. As projected from the prior solubilization experiments, the addition of 40% methanol to the initial proteome fraction resulted in higher sample recovery following the conclusion of the TME experiment, whereby the total soluble protein increased from 57.1% to 76.2%. This result was also confirmed by SDS PAGE (see Appendix A).

From Table 1, the methanol-containing sample required only 2.8 min to deplete the SDS below 100 ppm. Under identical operating conditions (250 mA constant current), it took nearly 8.8 min for the control sample (no methanol) to reach an equivalent level of SDS. Comparing the decay constants, the addition of methanol provided a 73% enhancement in the rate of SDS depletion in a constant-current TME experiment. This in turn translates to a smaller temperature increase, as the TME experiment operates for a shorter period of time. With methanol, the final temperature was 35 ± 3 °C, implying that the operating current could still be increased if a faster SDS depletion rate is desired. Previous work has noted a temperature near 60 °C as the upper limit above which protein aggregation significantly impacts yield [31].

We questioned whether the accelerated detergent depletion experienced from a constant-current experiment may be attributed to the increased resistance of the methanol-containing solution. As seen in Figure 2A, the solution resistance changes throughout the TME operation, causing the voltage to drop during a constant current run. The higher voltage for the methanol-containing sample can be explained by the higher resistance of the sample. As seen in Table 1, the resistivity of 40% methanol is 63% greater than that of water alone. It should be stated that the concentration of methanol in the TME experiment relates only to that in the sample cell and not to the cathode/anode buffers. Thus, a more direct comparison is provided by fixing the voltage throughout an SDS depletion run. Under these conditions, the drop in resistance forces the current to increase over time (Figure 2B), which translates to higher Joule heating, and a more dramatic temperature increase as the run progresses. Nonetheless, the addition of methanol continued to yield a higher rate of SDS depletion in a constant-voltage TME experiment. From Table 1, the decay constant increased from 0.99 ± 0.02 to 1.31 ± 0.07, a 32% gain, which cannot be attributed to a change in solution resistance. This accelerated rate of SDS depletion in methanol warranted further investigation to explain the observation.

### 3.2. Methanol Increases the CMC and Lowers the SDS to Protein Binding Energy

We hypothesized that a change in the CMC of the surfactant in methanol could explain the faster rate of SDS depletion in the TME experiment. As suggested by Kachuk et al., the size of detergent micelles impedes their flux through the MWCO membrane, implying that a higher concentration of free detergent monomers would increase electrophoretic flux [32]. As summarized in Table 2, from conductivity measurements, we confirmed the CMC of SDS in pure water to be 8.4 ± 0.1 mM (refer to Appendix A for conductivity graphs). This value closely aligned with the reported CMC of 8.3 mM in pure water [41]. From our conductivity measurements, in 40% methanol, the CMC increased threefold, to 26.5 ± 1.3 mM. Therefore, the inclusion of methanol causes a significant increase in the concentration of free detergent monomers, which would contribute to increasing the total flux of detergent through the membrane in the TME experiment.

Another potential variable that could explain the enhanced SDS depletion rate relates to the binding of SDS to protein in water vs. 40% methanol. From classic dialysis measurements with BSA, a ratio of 1.4 g SDS/g protein has been reported [44]. We repeated a comparable dialysis experiment and obtained a binding ratio of 1.4 ± 0.1 g SDS/g protein when prepared in an aqueous solvent. However, when dialyzed against 40% methanol, the SDS–protein binding ratio dropped to 0.30 ± 0.05 g SDS/g protein (Table 2). In other words, the presence of methanol lowers the binding affinity of detergents to proteins. This would again increase the free concentration of SDS monomer and contribute to a higher flux of detergent in the TME experiment.

To determine which variable (CMC or SDS to protein binding ratio) most significantly contributes to accelerating the SDS depletion rate in methanol, we performed a constant-current TME experiment in the absence of protein (Figure 3A,B). Under these conditions, the addition of methanol provided a 55% increase in the decay constant (Table 2). Recall a 73% enhancement in the rate of SDS depletion when protein was present. However, we found no significant difference when comparing these results (*p* = 0.18), suggesting the SDS–protein binding ratio had a minimal impact on the depletion rate. This was to be expected. As has been discussed previously, the applied electric field is more than sufficient to overcome the SDS-protein binding energy, given that the residual SDS at the conclusion of a TME experiment is far below the equilibrium binding ratio [32]. Thus, the increase in CMC observed in 40% methanol is likely a more significant contributor to the accelerated SDS depletion rate.

### 3.3. Mass Spectrometry Analysis of Samples Containing 40% Methanol vs. No Methanol

Following SDS depletion by TME, the soluble protein fractions were collected and subjected to bottom-up proteome analysis by mass spectrometry to characterize the membrane proteins recovered from the automated sample preparation workflow. A complete listing of identified proteins, together with the observed intensity ratios (methanol vs. water), is provided in Appendix A.

Figure 4A summarizes the observed overlap in detected proteins recovered from the aqueous sample vs. 40% methanol. As expected, we observed a high agreement in identified proteins, with 1481 unique protein components in common (86% overlap). Additionally, while only 25 proteins were unique to the control preparation (water), 222 proteins were uniquely detected in the methanol-containing sample.

The intensity ratios of proteins common to each fraction were extracted from the MS data and transformed to a log2 scale. The histogram of Figure 4B shows an expected Gaussian spread in the abundance ratio of distinct proteins, with an average log 2-fold change of 0.628, favoring higher abundance in the methanol preparation. Thus, from MS analysis, the recovery of proteins from the methanol-containing sample was on average 1.5 × higher than that of the aqueous sample. The volcano plot in Figure 4C isolates several proteins with abundance ratios deemed statistically significant. We selected a *p*-value of 0.1 (shown in the figure with a dashed line), allowing isolation of a subset of 68 proteins considered to have statistically significant abundance that was higher in the methanol-containing preparation, while 54 proteins were enriched in the water preparation (see Appendix A). These proteins were added to the set of uniquely identified proteins from the methanol and water preparations and used for the classification of proteomic trends.

We first classified the enriched MS-identified proteins according to their subcellular location, as shown in Figure 5. A detailed description of the gene ontology data for the enriched proteins is provided as Appendix A. Of the 290 proteins enriched in the methanol fraction (i.e., 222 uniquely identified plus 68 with a higher abundance ratio translating to a *p* value above 0.1), 147 were classified as membrane proteins (51%). Of these, 92 (63%) were classified as integral membrane proteins with at least 1 predicted transmembrane segment; GPI ethanolamine phosphate transferase was exclusively identified in the methanol preparation and had the greatest number of transmembrane segments with 15 (see Appendix A). This confirms the enrichment of membranes in this fraction. Interestingly, a similar percentage of proteins were also classified as membrane-associated in each of the other proteome subsets (Figure 5). Considering proteins enriched in the water preparation, 45% were also classified as membrane-associated. Furthermore, 19 of the 35 membrane proteins enriched in the water fraction were classified as integral membrane proteins (54%). The protein, high-affinity hexose transporter HXT7, had the greatest number of predicted transmembrane segments with 12; this protein was also observed in the methanol preparation, albeit with a lower abundance ratio than the water preparation (MeOH/Water = 0.372). Appendix A compares the distribution of membrane proteins according to the number of transmembrane segments. Again, a similar trend is revealed. The fraction of membrane proteins enriched in the methanol fraction is not statistically different from the water-enriched fraction, nor from the proteins with similar intensity ratios between fractions.

We compared various protein properties of proteins enriched in the methanol or water fractions, including their hydropathy index (GRAVY scores) and protein molecular weight trends (Figure 6). No significant differences were noted. However, considering the isoelectric points, proteins enriched in the methanol preparation showed a higher average pI (7.6 ± 1.8) than those elevated in the aqueous preparation (6.7 ± 1.7), providing a statistically significant difference with *p* < 0.05 (Figure 6C). This is further illustrated in Appendix A. Our SDS depletion experiments were conducted in a buffer at pH 8.1. Consequently, proteins with a higher abundance in the methanol extract are more likely to have a net charge closer to 0. This is shown in Figure 6C, as proteins enriched in the water sample have a higher absolute net charge. This observation aligns with the expected solubility of proteins in an aqueous solvent, where proteins tend towards minimal solubility as the pH approaches the pI. The inclusion of methanol during SDS depletion by TME therefore serves to minimize the aggregation of proteins with limited solubility in water, influenced by their reduced charge state.

## 4. Discussion

MS-compatible solubilizing additives have been suggested as a compromise between proteome solubilization efficiency and MS compatibility. The strategy outlined here initially employs SDS, which maximizes membrane proteome recovery [7], with the objective of retaining proteome solubility throughout the process of SDS depletion. This is accomplished by incorporating 40% methanol within our electrophoretic detergent removal platform, TME, ensuring that membrane proteins remain soluble as the surfactant is removed. As intact proteins are retained in solution throughout the purification process, the workflow is ultimately compatible with both top-down and bottom-up MS analysis [20].

It has previously been suggested that 60% methanol can be employed as an MS-compatible workflow for proteome solubilization and trypsin digestion ahead of bottom-up analysis [15,16]. To date, the approach has not been demonstrated for the preparation and analysis of intact membrane proteins. Furthermore, no prior study examined whether 60% methanol was optimal for membrane proteome extraction. Rather, 60% methanol was a logical solvent based on prior use to minimize interactions between membrane proteins and lipids [44]. It has also been acknowledged that trypsin activity is significantly tempered by 60% methanol [15]. Our group has previously demonstrated that the addition of trypsin to precipitated protein is still capable of generating peptides, despite minimal protein solubility [18]. A study by Moore et al. identified over 10 times more proteins (941 vs. 88) when comparing proteome sample preparation workflows involving MS-compatible surfactants to those involving 60% methanol [45]. The present study shows that 60% methanol will precipitate most proteins. Alternatively, 40% methanol maximizes protein solubility while retaining higher trypsin activity and providing a direct route to process the detergent-depleted fraction for bottom-up analysis. During SDS depletion by TME, we demonstrated a 33% increase in protein recovery with the inclusion of 40% methanol in the sample buffer (Table 1).

An additional objective of the TME workflow is to maximize the throughput of SDS depletion. Higher voltage will deplete SDS faster but also result in greater heat generation, which increases the likelihood of protein aggregation. As shown in Table 1 and Table 2, the addition of methanol to the TME buffer provided an enhancement in the rate of SDS depletion. This was rationalized by the resulting impact of methanol on raising the CMC of SDS. As highlighted in the Appendix A, 40% methanol increases the CMC of the surfactant approximately 3-fold, from 8.4 to 25.7 mM. Prior studies have confirmed an increase in the CMC of SDS, with some discrepancies based on the method of detection (e.g., conductivity vs. surface tension) and some lab-to-lab variances [46,47]. A trend of increasing the CMC with the addition of methanol serves to increase the rate of surfactant transport through the membrane.

We also examined the influence of SDS-protein binding in MeOH. The classic ratio of 1.4 g SDS per gram of protein drops substantially to 0.3 g SDS per g of protein in methanol. This could serve to enhance detergent depletion, noting that the strong electric field employed in TME is sufficient to overcome SDS-protein binding, implying that this is less impactful than the corresponding CMC increase in methanol. Nonetheless, the quantified drop in the SDS-protein binding ratio may prove insightful for other detergent depletion strategies, such as column chromatography or centrifugal membrane filters.

MS analysis of the recovered proteins following SDS depletion confirms the benefits of incorporating methanol into the depletion workflow. As expected, the intensity ratio of proteins recovered in the methanol fraction was generally higher than that of the water fraction. As seen in Figure 4, certain proteins were deemed to have statistically altered abundance. Our analysis concludes that methanol did not impact the recovery of proteins according to their molecular weight, meaning that 40% methanol will not inadvertently precipitate larger molecular-weight proteins. Interestingly, methanol did not appear to enhance the recovery of ‘hydrophobic’ proteins according to their GRAVY scores. This can be explained given that the addition of SDS for initial extraction serves to maximize the solubility of all protein types. Though ionic surfactants are incompatible with downstream MS analysis, their use to isolate and fractionate intact proteins and proteoforms serves as a favorable sample preparation strategy for top-down analysis. Thus, during TME depletion, the objective of including methanol is only to maintain protein solubility for a sample that is already free of lipids, or protein aggregates. The organic solvent provides the greatest benefit in maintaining the solubility of intact proteins with minimal net charge prior to subsequent digestion. The influence of charge on protein solubility is well documented. Thus, adding methanol serves to retain proteins most likely to aggregate during TME depletion and suggests that the charge state of the protein is a primary factor determining their solubility.

## 5. Conclusions

Membrane proteins are important in clinical studies but pose a greater challenge for detection given their lower solubility. Here, we demonstrate an SDS-based workflow to maximize proteome extraction efficiency, followed by detergent depletion through an automated electrophoretic platform termed transmembrane electrophoresis. We show that the addition of 40% methanol maximizes the solubility of a membrane proteome fraction and provides the added benefit of accelerating the rate of SDS depletion. We can therefore identify a greater number of SDS-purified proteins with greater MS signal intensity, particularly those exhibiting minimal net charge in the TME operating buffer. It is further recommended that organic solvent-based proteomic workflows consider lowering the amount of organic solvent, from 60 to 40% methanol, to minimize the risk of protein precipitation and therefore improve the analysis of membrane proteins.

## Figures and Tables

**Figure 1 proteomes-12-00005-f001:**
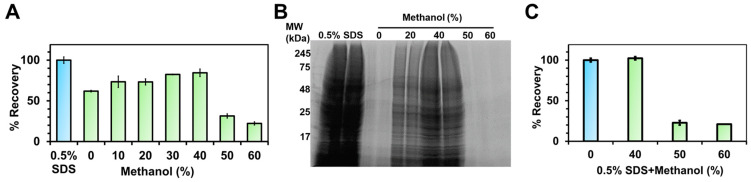
(**A**) Recovery of proteins extracted from a membrane preparation with increasing concentrations of methanol in water, relative to the protein recovery from the same sample obtained with 0.5% SDS in water. (**B**) SDS-PAGE image of the extracted membrane proteins recovered with varying solvents as indicated. (**C**) The solubility of membrane proteins, previously extracted with 0.5% SDS, following the addition of methanol to the final concentration listed. The recovery of proteins remaining in the solution was determined following incubation (30 min) and centrifugation of the sample. Error bars represent standard deviations, *n* = 3.

**Figure 2 proteomes-12-00005-f002:**
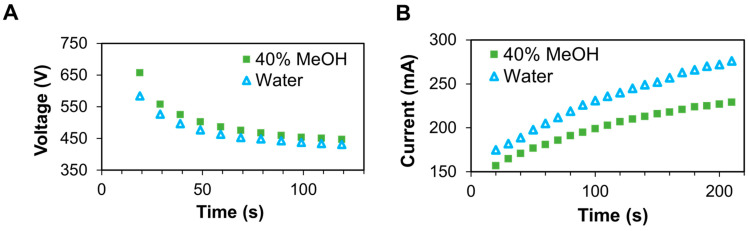
(**A**) Voltage plot of the TME experiment operated at a constant current of 250 mA; (**B**) Current plot of TME when operating at a constant voltage of 350 V.

**Figure 3 proteomes-12-00005-f003:**
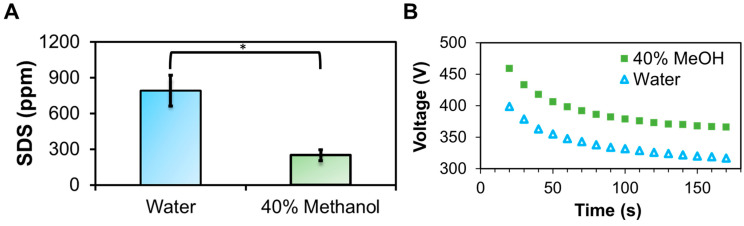
(**A**) Bar graph of residual SDS following TME operation at a constant current of 250 mA, operated in the absence of protein and initially containing 0.5% SDS. (**B**) Voltage plot over the 2 min runs. * Indicates a statistically significant difference at *p* < 0.05.

**Figure 4 proteomes-12-00005-f004:**
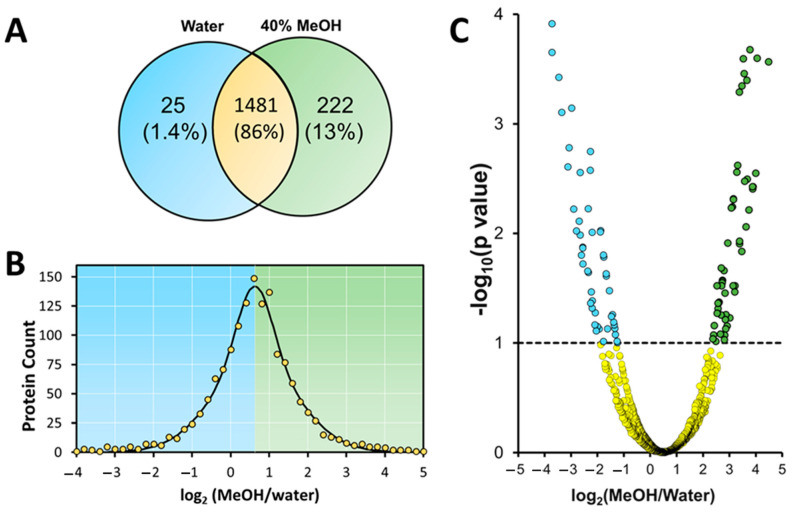
(**A**) Venn diagram of proteins identified from a membrane-enriched fraction processed by TME with 40% methanol and no methanol. (**B**) Histogram of log2 abundance ratios of proteins commonly identified in the 40% methanol and water fractions, showing the preference towards higher abundance in the methanol fraction. (**C**) Plot of −log10 of *p* value vs. log_2_ of the abundance ratio (methanol/water). The left side of the graph (blue dots) indicates proteins that were found to be significantly higher in the water preparation, while the right side (green dots) indicates proteins that are significantly higher in the 40% methanol preparation (yellow dots = unchanged proteins).

**Figure 5 proteomes-12-00005-f005:**
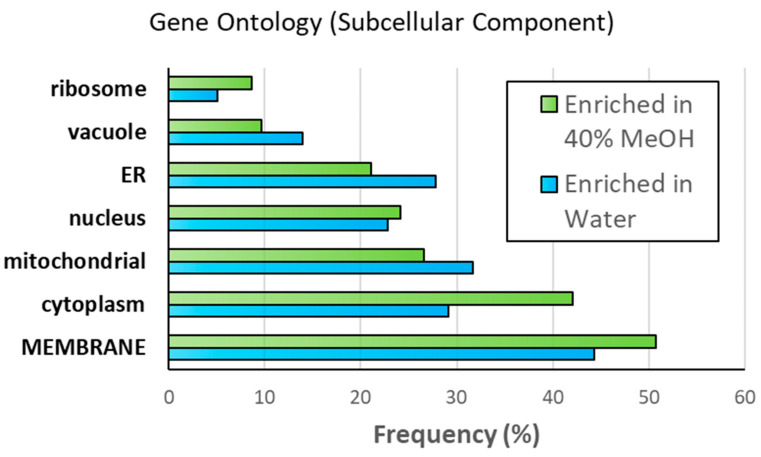
Sub-cellular localization of proteins classified as enriched in the 40% methanol preparation (290 total), or in the conventional water preparation of TME (79 total proteins).

**Figure 6 proteomes-12-00005-f006:**
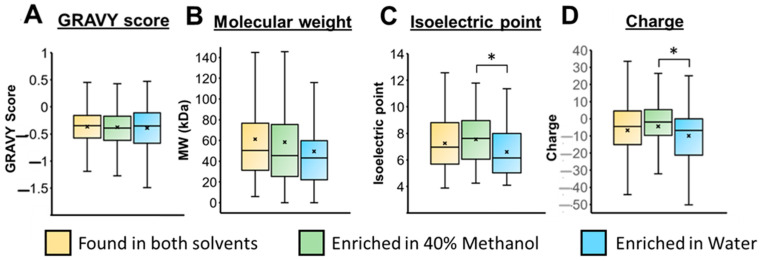
Box and whisker plots for proteins that were commonly identified in both solvents vs. those enriched in 40% methanol or enriched in the water fraction, summarizing (**A**) GRAVY scores; (**B**) molecular weight distributions; (**C**) protein isoelectric points; and (**D**) the net protein charge calculated at a pH of 8.1. * Indicates a statistically significant difference at *p* < 0.05.

**Table 1 proteomes-12-00005-t001:** SDS depletion from membrane proteome fraction at constant current in both water and 40% methanol.

Sample	Water	40% MeOH	Ratio (MeOH/H_2_O)	*p*-Value
Protein recovery (%)	57.1 ± 4 ^A^	76.2 ± 7	1.3 ± 0.2	0.02
Time to 100 ppm (min)	8.8 ± 0.9	2.8 ± 0.5	0.31 ± 0.1	5 × 10^−4^
Temperature (°C)	41 ± 4	35 ± 3	0.85 ± 0.1	0.11
Decay constant (min^−1^) at 250 mA	0.65 ± 0.001	1.13 ± 0.001	1.73 ± 0.2	6 × 10^−4^
Decay constant (min^−1^) at 350 V	0.99 ± 0.02	1.31 ± 0.1	1.32 ± 0.08	0.002
Resistivity (µS)	0.0015 ± 0.0002	0.0025 ± 0.001	1.63 ± 0.01	1 × 10^−5^

^A^ Average ± standard deviation, where *n* = 3.

**Table 2 proteomes-12-00005-t002:** Measured CMC of SDS and SDS–protein binding in water and 40% methanol. The standard deviation is taken from repeating the experiment 3 times for (*n* = 3).

Solvent	Water	40% Methanol	Ratio (MeOH/H_2_O)
CMC (mM)	8.4 ± 0.1 ^A^	25.7 ± 1.3	3.1 ± 0.2
SDS-protein binding (g/g)	1.4 ± 0.1	0.30 ± 0.05	0.2 ± 0.04
Decay constant with no proteins (min^−1^)	0.91 ± 0.02	1.41 ± 0.04	1.55 ± 0.06

^A^ Average ± standard deviation, where *n* = 3.

## Data Availability

A complete listing of identified MS-proteins together with intensity values is provided as Appendix A.

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
