# Peer review of "Enhanced Electrophoretic Depletion of Sodium Dodecyl Sulfate with Methanol for Membrane Proteome Analysis by Mass Spectrometry"

_proteomes, 2024, doi:10.3390/proteomes12010005_

Round 1

Reviewer 1 Report

Comments and Suggestions for Authors

The authors describe the use of 40% methanol as the optimal concentration during depletion of SDS from membrane proteins using transmembrane electrophoresis (TME). 

SDS is remarkably efficient at solubilizing membrane proteins typically used in electrophoretic applications. Removal of SDS for further processing of proteins is more challenging. The authors show here that addition of methanol to 40% is effective for this purpose when TME is used to remove SDS.

The paper is highly focused and well written. I have only minor points for the authors to consider.

SDS. As the authors point out SDS is an ionic detergent. Thus in solution it exists as the negatively charged dodecyl sulfate (DS-) with sodium ions. I suggest the paper would be improved by recognizing this and not referring to it as SDS the whole time. This complicates the situation a little as the competing presence of potassium ions for example would modify its behavior. 

The authors point out the potential to observe heating during TME. This has the potential to impact recovery of proteins. It is thus important to control this as variable recovery of some proteins would impact their ability to be quantified. 

Author Response

Reply to Reviewer 1

1) As the authors point out SDS is an ionic detergent. Thus in solution it exists as the negatively charged dodecyl sulfate (DS-) with sodium ions. I suggest the paper would be improved by recognizing this and not referring to it as SDS the whole time. This complicates the situation a little as the competing presence of potassium ions for example would modify its behavior.

This is a very valid point. Indeed, the dodecyl sulfate anion (DS-) represents the portion of SDS presenting the greatest challenge in being depleted from protein samples, as this is the component of the SDS which directly binds to protein and is not easily removed (Na+, by comparison, would be removed by reversed phase LC ahead of MS).  It is recognized that in an electrophoretic platform, both Na+ and DS- will be depleted (Na+ migrates to the cathode, DS- to the anode). Thus, to refer to TME as a platform for ‘SDS’ depletion remains valid (just that it is depleted in 2 parts). The presence of potassium ions would induce (partial) precipitation of DS- as a low solubility KDS salt.

We have revised the manuscript to clarify how SDS is isolated from proteins during the electrophoretic process, with DS-. This is found in the 3rd paragraph of the intro (page 2, line 78).

 2) The authors point out the potential to observe heating during TME. This has the potential to impact recovery of proteins. It is thus important to control this as variable recovery of some proteins would impact their ability to be quantified.

The reviewer is pointing to one of the most important limitations of TME, and one which is directly addressed in this work. Joule Heating, or resistance heating, is an unavoidable consequence of passing a current through a resistor (the sample solution).  The magnitude of Joule Heating, Q is a function of the current (I) applied to the sample solution, of given resistance,  R, (Q = I2 R). We control the magnitude of Joule Heating by operating TME at a maximum operating current (in this work, 250 mA). It is the reason why we can’t simply raise the voltage to deplete SDS faster – if so, protein yield would suffer. Here, we report both constant current, and constant voltage experiments (with vs without methanol), and discuss the implications of adding methanol on Joule Heating. 

The revised manuscript expands on the implications of Joule Heating in the 3rd paragraph of the introduction (page 2, lines 85-90). We also note the upper temperature limit (60oC) where protein recovery becomes most significantly impacted, as seen from prior work (line 329-330). Additional discussion of the importance of temperature is included in the manuscript.

Reviewer 2 Report

Comments and Suggestions for Authors

SDS is usually added when performing the membrane protein extraction. Given that SDS would interfere with the downstream analysis of MS, the authors here found that 40% methanol could also improve the solubility of membrane proteins, although which is already reported.

However, I didn’t sense the novelty. Each part can be more concise so that the readers can concentrate on your crucial points.

In the results 3.3, fig.5, does this manuscript talk about the membrane proteins? Based on your MS data, the largest part is only 45% membrane proteins.

Author Response

Reply to Reviewer 2

  • SDS is usually added when performing the membrane protein extraction. Given that SDS would interfere with the downstream analysis of MS, the authors here found that 40% methanol could also improve the solubility of membrane proteins, although which is already reported. However, I didn’t sense the novelty.

We note that prior studies consistently recommend 60% methanol to solubilize membrane proteins; here we showed that 60% methanol induces protein aggregation, lowering yield, and therefore demonstrate 40% methanol as the appropriate alternative. However, this alone is not the primary novelty of the work – rather it is to employ 40% methanol to maximize membrane proteome solubility during, and following SDS depletion by TME. Doing so not only improves recovery (as highlighted by the quantitative proteomics analysis), but further accelerates the rate of SDS depletion. In other words, we still incorporate the benefits of SDS for initial proteome extraction, but ensure that the protein solubility is retained in an MS-compatible solvent. The manuscript explores this in detail, explaining what contributes to the accelerated depletion rate.

  • Each part can be more concise so that the readers can concentrate on your crucial points.

We appreciate the reviewer’s comment and agree that the manuscript presented a lot of information which could be difficult to follow. As suggested, we have made substantial changes to reduce the overall length, particularly within the results and discussion sections, while ensuring that the most crucial points are emphasized.  Please refer to the highlighted changes of the revised manuscript.

  • In the results 3.3, fig.5, does this manuscript talk about the membrane proteins? Based on your MS data, the largest part is only 45% membrane proteins.

Proteome analysis was conducted on a membrane-enriched fraction of yeast, based on the work of Wu et al. Anal. Chim. Acta 2001 (reference 37 in revised manuscript). As summarized in Figure 5, this membrane-enriched fraction contributed to the MS detection of 1703 proteins in the methanol fraction, of which 45% were classified as membrane proteins.  The membrane-enriched fraction would not be expected to yield 100% membrane proteins.

We have edited the revised manuscript to ensure that we properly refer to the membrane “enriched” proteome fraction.  We have also provided an additional paragraph which summarizes the classification of the various identified proteins, beyond what was presented in Figure 5.

Reviewer 3 Report

Comments and Suggestions for Authors

Comments to the Authors

General comments:

In this manuscript, the authors proposed a new strategy for improving membrane protein enrichment and preparation for mass spectrometry (MS) analysis. This approach could be very useful for studying membrane proteins because it is simple, cheap and appears to be suitable for ESI-MS proteomic analysis. After an accurate re-reading of the paper, I suggest (see comments in “Major points”) improving the proteomics results in order to highlight the potentialities of the new approach in MS-based proteomics field.

For this reason, I think that the manuscript is not suitable for publication in this form. Some integrations and suggestions for improving the manuscript are provided below as specific comments.

Minor points:

·       Line 183: sufficient is right?

·       Lines 256-259: authors cited Kachuk et al., but reference 28 is reported (Crowell et al.), maybe it is wrong.

Major points:

·       Since this paper is submitted to the section “Proteomics Technology and Methodology Development”, I strongly believe that it is appropriate for the authors to create a nice and simple graphical abstract or experimental workflow to submit as main figure in order to better illustrate the developed methodological technique. This will make it easier for readers to follow the scientific work and redo it.

·       Method section is carefully described. However, both MS analysis and proteomics processing with Proteome Discoverer lack description of parameter settings. I would like to know more details about them. On the one hand, according to MS analysis, is the acquisition made in a single shot for the three samples or did the authors perform biological or technical replicates? On the other hand, for proteomics processing, I mean parameters of quantification (intensity or what?), MS and MS/MS tolerances, number of peptides used for quantification. At this time, I would ask to the authors why FDR was set to 5%, it seems pretty high, or am I wrong?

·       Figure 2 shows the difference in terms of voltage and current between TME experiments with 40% methanol Vs only water. Are these differences significant? What I want to know is whether the authors have performed technical replicates of the experiment and if so, how many.

·       “Mass Spectrometry Analysis of Samples Containing 40% Methanol Vs no methanol” paragraph in the results section lacks much information. It is poor and not-well argued. I think that it should be the beating heart of the work, if the authors want to demonstrate the high capabilities of this new method in analyzing and studying membrane proteins through MS analysis. I suggest the authors clarify the accordance between the Venn Diagram (numbers of quantified proteins) and the proteins reported in Table S1: maybe the protein numbers do not match. Moreover, I would like to know which are “the 292 proteins enriched in the methanol fraction” (line 326) used for GO functional annotation. In this context, I suggest adding a specific paragraph where the GO analysis should be well described according to the quantified membrane proteins in order to use proteomics approach as valid and unbased technique to evaluate the robustness and MS-compatibility of the new method here presented. Which category of membrane proteins is well quantified by adding 40% methanol in sample preparation?

Author Response

Reply to Reviewer 3

Minor points:

1)      Line 183: sufficient is right?

Thank you for finding this typo – although in the revised manuscript, this phrase was removed as part of our overall efforts to reduce the length of the manuscript

2)      Lines 256-259: authors cited Kachuk et al., but reference 28 is reported (Crowell et al.), maybe it is wrong.

Another good catch!  We meant to cite reference 30 (Kachuck et al., J. Prot. Res., 2016) and have corrected the mistake… The incorporation of additional references has shifted the references in the revised manuscript.

Major points:

  • Since this paper is submitted to the section “Proteomics Technology and Methodology Development”, I strongly believe that it is appropriate for the authors to create a nice and simple graphical abstract or experimental workflow to submit as main figure in order to better illustrate the developed methodological technique. This will make it easier for readers to follow the scientific work and redo it.

It may be possible that our originally prepared graphical abstract (below) was not provided to the reviewer, and/or failed to upload properly. We have appended this image to the end of the revised manuscript, to ensure that it is included for reviewers. If it is felt that another figure should be generated, demonstrating our experimental workflow, we would be happy to prepare one.  The graphical abstract is meant to illustrate the solubilization of proteins from a membrane preparation with SDS as a first step, followed by addition of methanol, then removal of SDS (TME). The inclusion of methanol accelerates the rate of SDS removal, while quantitative proteomics analysis by MS confirms an increase in the abundance of proteins when methanol is included.

4)      Method section is carefully described. However, both MS analysis and proteomics processing with Proteome Discoverer lack description of parameter settings. I would like to know more details about them. On the one hand, according to MS analysis, is the acquisition made in a single shot for the three samples or did the authors perform biological or technical replicates? On the other hand, for proteomics processing, I mean parameters of quantification (intensity or what?), MS and MS/MS tolerances, number of peptides used for quantification. At this time, I would ask to the authors why FDR was set to 5%, it seems pretty high, or am I wrong?

To address these comments directly, we did not perform any biological replicates, as the focus of the study was to highlight the technical capability of our novel analytical workflow. 2 independently prepared samples were processed by TME containing 40% methanol, while 2 more independently processed samples from TME were prepared in water. MS was conducted on each of these 4 samples.

 In addition to the above details (lines 205-208), we have also expanded the experimental section to include additional parametric settings for Proteome Discoverer (lines 219-222).

 FDR was set to 5% so as not to filter the detection of peptides from low abundance peptides. In other words, this is what contributes to the large percentage of overlap between the methanol and water fractions. The lower FDR is taken into account when calculating the confidence (p value) in the log2 (Fold Change) abundance ratios (figure 4C).

5) Figure 2 shows the difference in terms of voltage and current between TME experiments with 40% methanol Vs only water. Are these differences significant? What I want to know is whether the authors have performed technical replicates of the experiment and if so, how many.

The data shown in Figure 2 were not averaged; they represent a single run, noting that 5 samples are simultaneously processed in a single TME run; we always refer to the total current applied across all samples. Our device has multiple same cell chambers, as has previously been reported. However, these experiments have been repeated multiple times, and we consistently observe the trends reported.

6)  “Mass Spectrometry Analysis of Samples Containing 40% Methanol Vs no methanol” paragraph in the results section lacks much information. It is poor and not-well argued. I think that it should be the beating heart of the work, if the authors want to demonstrate the high capabilities of this new method in analyzing and studying membrane proteins through MS analysis. I suggest the authors clarify the accordance between the Venn Diagram (numbers of quantified proteins) and the proteins reported in Table S1: maybe the protein numbers do not match.

We have clarified the number of proteins summarized in the Venn diagram relative to those reported in Table S1. Indeed, there was an error; 222 proteins were uniquely detected in the methanol preparation, and 1481 were common to both. The pie charts shown in Figure 5 were generated based on the correct lists of proteins, though we have corrected the total counts of proteins from these lists (percentages do not change). We have also added additional worksheets to the excel file containing Table S1, so as to present subsets beyond (1) the original total list of proteins, together with (2) proteins unique to methanol (3) unique to water (4) enriched in methanol, and (5) enriched in water.

7) Moreover, I would like to know which are “the 292 proteins enriched in the methanol fraction” (line 326) used for GO functional annotation.

As discussed above, we have corrected the text to refer to 290 proteins enriched in the methanol fraction, referring to the combination of 222 proteins unique to the methanol fraction plus 68 wherein the log2(Fold Change) was statistically significant (green dots of Figure 4C). A worksheet containing this subset is provided in the supplement. We have ensured these number are clearly described in the revised manuscript (line 444).

8) In this context, I suggest adding a specific paragraph where the GO analysis should be well described according to the quantified membrane proteins in order to use proteomics approach as valid and unbased technique to evaluate the robustness and MS-compatibility of the new method here presented. Which category of membrane proteins is well quantified by adding 40% methanol in sample preparation?

 Figure 5 summarized the sub-cellular localization of identified proteins, stressing if proteins were classified as part of the membrane. We recognized the different classifications of membrane proteins, and therefore expand on this in the revised manuscript. We include a supplemental table (S2) which provides the gene ontology data. In the revised manuscript, we discuss the number of transmembrane segments for integral membrane (lines 447-466) and provide an additional Figure (Supplemental Figure S4) which helps visualize this data.    Figure 5 was also revised slightly to reflect this latest analysis of the classification of proteins, noting that the conclusions remain the same – proteomics data confirms the detection of a large fraction of membrane proteins, from both the methanol and water enriched fractions.

Reviewer 4 Report

Comments and Suggestions for Authors

The article proposed by the authors is based on an improvement in techniques to increase the identification of membrane proteins using traditional proteomics techniques. Highly hydrophobic membrane proteins are important in clinical trials, but their detection is more difficult due to their low solubility. The work presented in this article is based on the use of SDS to maximise the efficiency of proteome extraction, followed by detergent depletion using technology based on transmembrane electrophoresis. The results clearly show that the addition of 40% methanol maximises the solubility of a membrane proteome fraction and has the advantage of accelerating the rate of SDS depletion. Two comments should be made which do not call the work into question. First, yeast is a relatively simple unicellular organism. Did the authors try to tackle more complex systems or even tissues? If so, would the results be the same? Second, proteomics has traditionally used SDS-PAGE separation. For broad coverage of the proteome, especially membrane proteomes, stacking gel separation is particularly recommended. Have the authors compared the two approaches? If so, did they observe a greater number of identifications and a greater coverage of sequences? Without calling their results into question, a comparison could be made before publication to corroborate their findings.

Author Response

Reply to Reviewer 4

Two comments should be made which do not call the work into question.

  • Yeast is a relatively simple unicellular organism. Did the authors try to tackle more complex systems or even tissues? If so, would the results be the same?

Though yeast is a simple model system, proteomics analysis of the enriched membrane proteome fraction identified over 1700 unique proteins.  More powerful MS instrumentation would likely increase these numbers, though comparative analysis of sample preparation conditions (TME with 40% methanol or water) employed identical MS detection, making the comparison valid.

In our work, the processing of the proteomic system by TME takes place on a sample wherein the proteome has already been isolated. Inclusion of SDS enhances the initial extraction of proteins. To extract a tissue sample (eg we’ve worked with plant seeds, E. coli, salmon tissue, bovine liver, animal cell lines, exosomal purifications) may require additional efforts to maximize proteome recovery (sample homogenization, agitation during extraction, application of heat, etc…) However, once extracted, TME processing exhibits identical behaviour.

  • Proteomics has traditionally used SDS-PAGE separation. For broad coverage of the proteome, especially membrane proteomes, stacking gel separation is particularly recommended. Have the authors compared the two approaches? If so, did they observe a greater number of identifications and a greater coverage of sequences? Without calling their results into question, a comparison could be made before publication to corroborate their findings.

Thank you for the suggestion.  We acknowledge the potential for increased proteome identification following additional sample processing. Our group previously introduced the GELFrEE technique, which fractionates proteins according to molecular weight, but avoids the excision of proteins from SDS PAGE gels.  We recently completed a study comparing in-solution digestion, detergent-based sample preparation, and conventional (short GeLC) in-gel digestion. In fact, in-gel digestion outperforms in-solution digestion both in total number of proteins identified, and quantitative precision. The detergent-based approach (without gels) proved superior to gel-based approaches. These results are to be reported elsewhere (manuscript accepted with minor revisions).

The emphasis of this study is to demonstrate the benefits of incorporating methanol into a detergent-based platform for membrane proteome analysis.  Once the detergent is depleted, retaining proteome solubility can be challenging, and this is where the benefits of 40% methanol can be realized. We also demonstrated the accelerated rate of SDS depletion on inclusion of methanol.  As our comparative proteomics analysis analyzed the samples under identically controlled conditions, we did not include additional preparative steps. In fact, the more preparation steps are incorporated, the greater the analytical variance. Thus, a direct analysis of the purified samples provides the most valid data to compare.

Round 2

Reviewer 3 Report

Comments and Suggestions for Authors

•    The authors replied to the request to classify membrane proteins by uploading supplementary Table S2 highlighting that the addition of 40% methanol improve the membrane proteins yields. In this context, the current Figure 5 is clearer than previous one. In general, the authors better emphasize the novelty of the application that is well described in graphical abstract. Now is suitable for publication.

•    A little error: Venn Diagram reports 86% of overlap, instead in the text it is 88%. In general, are the percentages shown in the Venn diagram correct? 

Author Response

Thank you for finding this error. The % overlap works out to 86% (1481 out of 1728 total proteins).

We have also revised the % overlap in the Venn diagram of Figure 4... 25 proteins out of 1728 works out to 1.447%.  Very close to 1.5%, though the number should technically have been rounded down to 1.4%.